# Bronchoscopic Features and Morphology of Endobronchial Tuberculosis: A Malaysian Tertiary Hospital Experience

**DOI:** 10.3390/jcm11030676

**Published:** 2022-01-28

**Authors:** Nurul Yaqeen Mohd Esa, Siti Kamariah Othman, Mohd Arif Mohd Zim, Tengku Saifudin Tengku Ismail, Ahmad Izuanuddin Ismail

**Affiliations:** 1Sunway Medical Centre Velocity, Kuala Lumpur 55100, Malaysia; 2Gleneagles Medini, Nusajaya 79250, Malaysia; dr_pegaga@yahoo.com; 3Faculty of Medicine, Universiti Teknologi MARA (UiTM), Shah Alam 40450, Malaysia; firadhom@yahoo.com (M.A.M.Z.); ahmadizuanuddin@gmail.com (A.I.I.); 4Medical Department, KPJ Tawakal Hospital, Kuala Lumpur 53000, Malaysia; tengkusaifudin71@gmail.com

**Keywords:** endobronchial tuberculosis, bronchoscopy, morphology, Malaysian

## Abstract

The diagnosis of endobronchial tuberculosis (EBTB) is difficult as it is not well visualized radiologically, and bronchoscopy is not routinely performed for tuberculosis (TB) patients. Bronchoscopic characterization via endoscopic macroscopic features can speed up the diagnosis of EBTB and prompt immediate treatment. In this study, we identified the clinical and bronchoscopic morphology of 17 patients who were diagnosed with EBTB from 2018 to 2020. Demographics, radiological, microbiological and histopathological data were recorded. Endobronchial lesions were classified according to Chung classification. The diagnosis was made based on a histopathological examination (HPE) of endobronchial biopsy, and/or positive ‘Acid-fast bacilli’ (AFB) microscopy/Mycobacterium tuberculosis (MTB) culture on microbiological examination of bronchial alveolar lavage (BAL) and/or positive MTB culture on endobronchial biopsy specimens. Furthermore, EBTB was predominant in young women, age 20 to 49 years old, with a male to female ratio of 1 to 2. Underlying comorbidities were found in 53% of the patients. Cough, fever and weight loss were the main symptoms (23.5%). The indications for bronchoscopy are smear-negative TB and persistent consolidation on chest radiographs. Consolidation was the main radiological finding (53%). An active caseating lesion was the main EBTB endobronchial subtype (53%). The leading HPE finding was caseating granulomatous inflammation (47%). All patients showed good clinical response to TB treatment. Repeated bronchoscopy in six patients post TB treatment showed a complete resolution of the endobronchial lesion. EBTB bronchoscopic characterization is paramount to ensure correct diagnosis, immediate treatment and to prevent complication.

## 1. Introduction

Endobronchial tuberculosis (EBTB) refers to a tuberculous infection of the tracheobronchial tree with microbial and histopathological features, with or without the involvement of the lung parenchyma [1]. It is a rare entity and often underdiagnosed as the symptoms are mainly attributed to pulmonary tuberculosis [2]. Radiological imaging and bronchoscopy are paramount for EBTB evaluation. Anti-tuberculous agent and prevention of airway stenosis are important for treatment [2,3,4]. Corticosteroids may halt the progression of active disease to fibro-stenotic stage albeit its role is still controversial. However, if complications such as post-obstructive pneumonia, atelectasis and dyspnea occurred due to tracheobronchial stenosis, airway patency must be reestablished mechanically by surgery or bronchoscopic methods [1].

The definite incidence of EBTB is under reported, as bronchoscopy is not part of routine examination for all TB patients. Hence, the percentage of endobronchial involvement in active pulmonary tuberculosis varies from centre to centre [5,6]. Chung et al. reported that EBTB present in 5.88% of pulmonary tuberculosis cases [4], while Kashyap et al. reported EBTB as 10–40% of their PTB cases [7]. This ratio was shown to be 50% in another two studies [8,9]. EBTB may mimic diseases such as asthma, pneumonia and lung cancer due to similarities of their symptoms [10,11]. Any site of the tracheobronchial tree may be affected by EBTB. It may lead to lung collapse, if it affects the middle lobe, since the entry to the middle lobe is small [12,13]. Chung categorized types of EBTB into the following seven subtypes based on bronchoscopic findings: edematous-hyperemic, actively caseating, fibrostenotic, tumorous, ulcerative, granular, and nonspecific bronchitic [14]. EBTB can be fatal with high bacilli loads, and may cause severe complications with high morbidity such as bronchial stenosis. Early diagnosis and treatment is therefore crucial [1,4,9]. The nature and outcome of EBTB varies, ranging from total cure to severe bronchostenosis [15]. In this study, clinical, radiological and bronchoscopic features of 17 EBTB patients in our centre were evaluated. This is, to our knowledge, the first study on EBTB bronchoscopic morphology recognition in a Malaysian cohort. This study will also reinforce on the importance of bronchoscopy among suspected TB patients. Appropriate treatment can be initiated immediately upon early diagnosis to prevent major complications. 

## 2. Patients and Methods

Two hundred and seventeen patients with chronic cough, three negative sputum microscopic examination for acid-fast bacilli (AFB) and/or persistent consolidation on chest radiograph (CXR) who were subjected and agreed for fibre-optic flexible bronchoscopy in Respiratory Unit, Hospital Selayang from 2018 to 2020, were examined. Persistent consolidation was defined as non-resolving consolidation after six weeks of treatment, despite completion of antibiotics. Seventeen of 217 AFB smear negative patients were found to have endobronchial lesion via bronchoscopic examination, and were included in the study. These 17 patients were diagnosed as having EBTB based on the histopathological examination (HPE) of endobronchial biopsy which showed granulomatous inflammation with caseation necrosis, and/or positive AFB-culture on the microbiological examination of bronchoalveolar lavage (BAL) and endobronchial biopsy. 

BAL-sample collection in our centre involves the instillation of sterile normal saline into a subsegment of the lung, followed by suction and collection of the instilled normal saline for analysis. This procedure is facilitated by the introduction of a flexible bronchoscope into a sub-segment of the lung. The BAL samples obtained were then sent for AFB microscopic examination and Mycobacterium tuberculosis (MTB) culture. For BAL MTB culture, ‘Lowenstein-Jensen’ egg-based solid media were used during the study period. This is a retrospective observational descriptive study which was done in accordance with the Helsinki declaration 2008. All patients gave their written informed consents. Age, sex, symptoms, co-morbidities, microbiological examination results, radiological findings on postero-anterior chest radiographs and bronchoscopic findings of the cases were recorded. 

Bronchoscopic lesions were categorized according to Chung classification [2]. Type 1 (actively caseating), type 2 (oedematous hyperemic), type 3 (fibrostenotic), type 4 (tumorous), type 5 (granular), type 6 (ulcerative) and type 7 (non-specific bronchitic). After diagnosis establishment, cases were treated with anti-TB medications: 5 mg/kg·day Isoniazid, 10 mg/kg·day Rifampicin, 20 mg/kg·day Ethambutol and 25 mg/kg·day Pyrazinamide. The treatment’s effectiveness was assessed with symptoms evaluation and repeated chest radiographs (CXR). Patients who had reverted to AFB culture negative and exhibited signs of successful treatment had continued treatment with a two-drug combination of Rifampicin and Isoniazid during the maintenance phase after completing two months of intensive anti-TB therapy. Anti-TB treatment was discontinued once patients had completed a 6-month course of treatment and were pronounced cured. However, due to their delayed therapeutic success, some patients required prolonged anti-TB treatment. Bronchoscopy was performed again to evaluate the patient’s treatment progress, the condition of earlier endobronchial lesions, treatment response, and any post-EBTB complications such as endobronchial stenosis. During the study period, none of the patients had a chest CT for the examination of lung infiltration because their repeated CXR demonstrated satisfactory resolutions. 

## 3. Results

Table 1 and Appendix A detail the patients’ age, gender, co-morbidities, symptoms, and chest radiograph findings. Females were more likely to have EBTB (M:F ratio: 1:2).

Over half of the EBTB patients were between the ages of 20 and 49. (10 cases, 59%).

In 53% (9 cases) of the patients, underlying comorbidities such as diabetes mellitus, HIV positive status, and others were discovered. Cough, fever with weight loss (4 cases, 23.5%), cough with hemoptysis (3 cases, 17.7%), and cough with fever were the most common symptoms (4 cases, 23.5%). Consolidations (9 cases, 53%) were the most common chest radiological findings, followed by consolidations with cavitary lesions (3 cases, 17.7%), consolidation with pleural effusion (2 cases, 11.8%), consolidation with lung mass (1 case, 5.8%), and others (2 cases, 11.8%). Localization of lung lesions from chest radiograph mainly involved right upper lobe (RUL) (5 cases, 29.3%), followed by right middle lobe (RML) combined with other segments (3 cases, 17.7%) and left upper lobe (3 cases, 17.7%).

Table 2 and Appendix A provide bronchoscopic characterization of endobronchial lesions from a macroscopic endoscopic perspective. Bronchoscopically, active caseating lesions (Type 1, 9 cases, 53%) were the most common endobronchial lesions seen (Figure 1).

Figure 2 shows edematous hyperaemic lesions (Type 2) which were seen in three cases (17.7%). Other EBTB subtypes such as tumorous lesions (Type 4) were found in two cases (11.8%), granular lesions (Type 5) in two cases (11.8%), and ulcerative lesions (Type 6) in one case (5.8%).

RUL (3 cases, 17.7%), RML (2 cases, 11.8%), and RLL (2 cases, 11.8%) were the most common bronchoscopy lesion localizations (2 cases, 11.8%). In four cases, the left lobe was involved (23.5%). Our patients’ CXR localizations and bronchoscopic findings showed no disagreement. (Appendix A). Caseating granulomatous inflammation was the most prevalent histopathological examination (HPE) of biopsy specimens, seen in 8 cases (47%), followed by granulomatous inflammation in 7 cases (41%), and chronic inflammation in 2 cases (12 percent). BAL for AFB microscopy was positive in 35% (6 cases), whereas BAL for Mycobacterium cultures was positive in 41% (7 cases) of the patients (Table 2). 

The majority of the patients (16 cases, 94%) achieved cure, based on clinical and radiological assessment, while 1 non-TB death occurred due to fluid overload in an EBTB patient with underlying End Stage Renal Failure (ESRF) after 2 months of anti-TB treatment. Table 3 and Appendix A summarise the findings. 

Bronchoscopies were repeated at the end of treatment in six cases (35%) where the endobronchial lesion had disappeared, while the remaining eleven patients rejected for personal reasons. Ten patients (59%) received successful therapy after six months of normal anti-tuberculous (TB) treatment, whereas four patients (23.5%) required nine months of anti-TB treatment. One patient (5.9%) required seven months of treatment, while another (5.9%) required a year of anti-TB treatment. Patients with underlying co-morbidities such as diabetes, cerebral palsy, and HIV were among the four patients who required nine months of anti-TB medication. One patient with underlying DM and rectal cancer required seven months of anti-TB medication. Another patient who required a year of anti-TB treatment was a post-kidney transplant patient who was on steroid and immunosuppressive medication. (Appendix A). Except for two patients, all 10 patients who recovered completely after six months of anti-TB treatment had no co-morbidities. One of them has SLE and is not on any steroid medicines, while the other is HIV positive. Because none of our EBTB patients developed a post-EBTB problem such as airway stenosis, no interventional pulmonology procedure was required (Appendix A). 

## 4. Discussion

Richard Morton identified EBTB in 1698 through an autopsy instance of tuberculosis mortality [3]. The pathophysiology of EBTB has yet to be determined. However, four probable explanations for the development of EBTB have been proposed: (I) A nearby parenchymal focus that results in direct invasion; (II) Infected sputum that results in direct implantation of the organisms; (III) Lymph node attrition within a bronchus; and (IV) lymphatic spread [4,5,6,7].

Early detection and treatment of tuberculosis and its complications, such as cicatricial bronchostenosis due to endobronchial involvement, can prevent the spread of tuberculosis and its complications [16]. Early diagnosis can be achieved by routine bronchoscopic assessment amongst PTB patients. More than half of the EBTB cases reported were in people under the age of 35. Variable female-to-male ratios have been described. It was noted that EBTB is more common in young women than in males [4,15,17]. This finding is consistent with our study, which showed a male/female ratio of M/F: 1/2. It was also noted to be common in the elderly, as described in another study by An JY et al. [8]. Different sites of tracheobronchial involvement were also reported in young and elderly populations [18]. Lobar and segmental bronchial involvement were commonly seen among elderly patients, while trachea and main bronchial involvement were commonly seen in younger patients. Middle-lobe syndrome was also noted to be more common among the elderly [19]. In our study, four out of five patients with middle lobe syndrome were females, and the majority of their ages were more than 40 years old (Appendix A).

EBTB symptoms include cough, very viscous sputum, wheezing, fever, pleuritic chest discomfort, and hemoptysis [16]. However, because these characteristics are also present in other lung disorders, they are ineffective in detecting EBTB early. The most common symptom of endobronchial inflammation is a persistent cough [16]. Cough, sputum, dyspnea, and fever were the most prevalent symptoms described by Qingliang X et al. in their study [20]. These findings are in line with our research, which found that cough, fever, weight loss, hemoptysis, and dyspnea were among the most common symptoms. 

The first test to diagnose PTB is sputum microscopy for AFB. However, the diagnostic yield for AFB microscopy positivity in EBTB is low, ranging from 16 to 53.3%, in contrast to Lee et al., who observed that sputum AFB microscopy positivity prior to bronchoscopy was as low as 17% [21,22,23]. The diagnostic yield rose to 73.6% with Mycobacterium culture [7]. To begin with, all of our EBTB cases exhibited negative sputum AFB microscopy, indicating the need for bronchoscopy in those individuals. Bronchoscopic lavage (BAL) exhibited positive AFB microscopy in six cases, positive Mycobacterium culture in eleven cases, and positive AFB microscopy and Mycobacterium culture in four cases, according to our findings (Appendix A). 

EBTB may have various radiological findings, ranging from patchy alveolar infiltrates, atelectasis, broad hilar, pleural effusion, mass, and cavitary lesions [16]. However, EBTB could not be ruled out in the case of a normal chest X-ray as 10% to 20% of EBTB cases may have a normal chest X-ray [16]. In Lee et al.’s report, the common radiological findings of EBTB were consolidations and volume loss, which constitute 83.4% of their cases [21]. Atelectasis is one of the most frequent chest X-ray findings of EBTB. Hence, it is challenging to differentiate EBTB from asthma and bronchogenic carcinoma in the elderly [21]. CT Thorax can provide more information in terms of demonstrating the presence of endobronchial masses and enlarged mediastinal-hilar lymphadenopathies. It also helps with a better description of lung parenchymal lesions and the assessment of complications such as stenosis or obstruction [2,16]. Qingliang X et al. described multiple lobar lesions, exudative shadows, and atelectasis as the most frequent radiological findings in their study [20]. The most frequent radiological findings in our study, listed according to their frequency, were consolidations, cavitary lesions, and mediastinal widening. 

Bronchoscopic evaluation is indicated in suspicious cases such as unexplained cough, wheezing, dyspnea, or haemoptysis [22]. Bronchoscopic evaluation should also be performed in cases with persistent segmental or lobar collapse, lobar infiltrations, or obstructive pneumonia findings on chest X-ray [7]. Lee JH et al. described that the right upper lobe and right main bronchus were amongst the most commonly diagnosed sites with EBTB via bronchoscopy [21]. In our study, common bronchoscopic localizations were in the right upper lobe (RUL), right middle lobe (RML) and right lower lobe (RLL). In a study by Park EJ et al. [17], the most common type was reported to be the caseous type. This is in agreement with our study, which also found the caseous type as the most common bronchoscopic finding amongst our EBTB patients. Another study reported on the edematous-hyperemic type as the most common bronchoscopic finding of EBTB [8,23]. In our study, the edematous-hyperemic type is the second-most common type of bronchoscopic findings amongst our EBTB patients. The caseous type is also associated with positive sputum and bronchial lavage AFB microscopy. Sputum and bronchial lavage AFB microscopy is usually negative in edematous type EBTB, hence the difficulty in diagnosis. Therefore, endobronchial biopsy for tuberculosis culture and histopathological examinations are indicated [19]. This is also in-line with our study, which found that 6 out of 9 cases of caseous type EBTB have either positive BAL AFB microscopy or positive BAL Mycobacterium culture. Caseous type EBTB is associated with a complete cure. In more than 60% of cases, the edematous or mixed caseous-edematous type converts to the fibrostenotic type [4].Variable degrees of luminal narrowing of the bronchus were observed in four of the subtypes, which are caseating, edematous-hyperemic, fibrostenotic, and tumorous EBTB. These findings were not found in the other three subtypes, which are granular, ulcerative, and nonspecific bronchitic EBTB [4].

These findings were also observed by Yanardag H et al., who reported that early stage exudative, granular, and ulcerative lesions recovered without complicated sequelae, while caseous and tumorous lesions of advanced disease might lead to complicated issues such as bronchostenosis associated with bronchectasia [16]. Um SW et al. discovered that age, more than 45 years old, pure or combined fibrostenotic subtype, and symptoms duration of more than 90 days prior to treatment were risk factors for bronchial stenosis complications [15]. In our study, the most common bronchoscopic lesions observed were active caseous lesions, followed by edematous-hyperemic, tumorous, granular, and ulcerative lesions with decreasing frequencies. Treatment responses in our cases were assessed via clinical symptoms, radiological and bronchoscopic improvement. There are studies suggesting that follow-up with CT can be an option instead of follow-up with bronchoscopy [16]. There is also literature mentioning that findings pertaining to bronchial stenosis development may also be shown via CT examination [16]. In our study, six patients who underwent bronchoscopic assessment post treatment showed features of successful treatment without any complications. Two cases were of the caseous subtype, three were of the tumorous subtype, and one was of the granular subtype. 

The use of corticosteroids in the prevention of fibrosis in tuberculosis patients has been studied. The use of steroids in the therapy of EBTB is, however, debatable [24,25]. Steroids had little effect on clinical improvement in EBTB patients, according to a few studies [5,6,26]. Several previous studies [2,27,28] found that oral or inhaled steroids promote clinical recovery in various kinds of EBTB. Corticosteroids are likely to be beneficial in the early stages, when hypersensitivity is the primary cause. Steroids, on the other hand, are no longer useful in more advanced cases where significant fibrosis is present. Close monitoring is advised since stenosis can occur later despite anti-tuberculosis treatment with or without corticosteroids [21,29,30,31,32,33,34,35,36,37]. There were 17 patients in the trial.

One patient died, and 16 of them were cured with standard anti-tuberculosis medication. Ten of them received the standard six-month treatment (intensive phase for the first two months, followed by four months of the maintenance phase). Multiple co-morbidities led to a delayed response to treatment, necessitating extended treatment of nine months in four cases, seven months in one case, and one year in one case. No oral corticosteroids were provided in any of our subjects since no indication of bronchial stenosis was seen. 

## 5. Conclusions

EBTB can present with a variety of clinical, radiological, and bronchoscopic features, some of which are unusual. They can also be mistaken for lung cancer, pneumonia, abscess, and other lung diseases. Females are more likely to develop them, and coughing is the most prevalent symptom. Diagnosing EBTB is difficult, which is why bronchoscopy is so critical. The most prevalent bronchoscopic subtype of EBTB is caseation, which affects the right lung lobes. Anti-tuberculous medications can be used to treat EBTB. Patients with co-morbidities, on the other hand, have a longer treatment time due to delayed therapeutic results. To avoid significant issues and ensure total recovery, it is critical to recognise the EBTB pattern and treat it right away. The acknowledgment of EBTB subtypes will also allow doctors to speak about EBTB using the same terminology. 

## Figures and Tables

**Figure 1 jcm-11-00676-f001:**
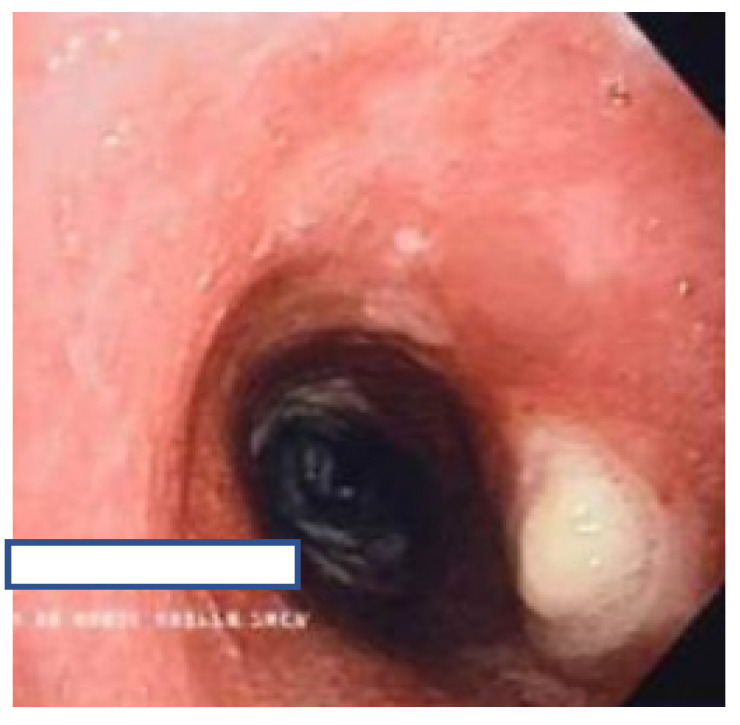
Caseating lesion. Type 1 Endobronchial TB.

**Figure 2 jcm-11-00676-f002:**
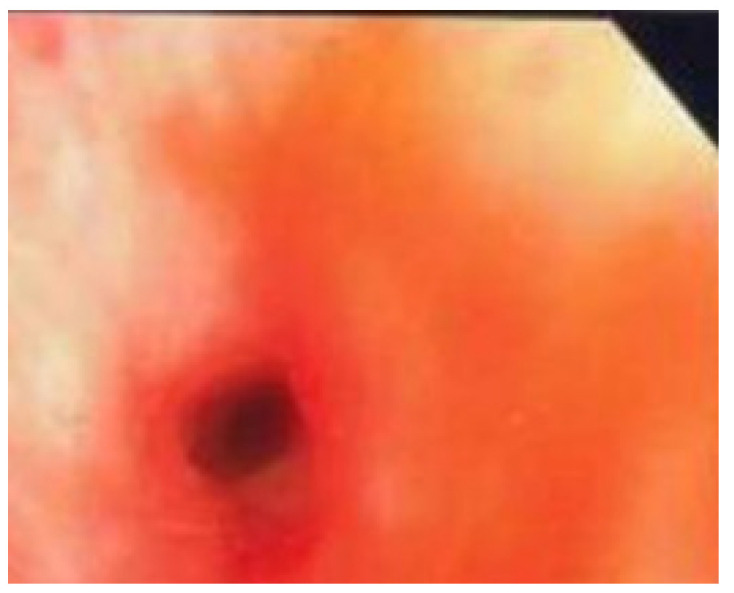
Oedematous hyperaemic lesion. Type 2 Endobronchial TB.

**Table 1 jcm-11-00676-t001:** Demographics and clinical data of EBTB patients.

Demographics	*n*	%
A. Gender		
Male	6	35.0
Female	11	65.0
B. Age (years old)		
<20	1	5.9
20–49	10	59.0
50–70	5	29.2
>70	1	5.9
Clinical features		
A. Co-morbidities		
None	8	47.0
Diabetes mellitus	4	23.5
Human immunodeficiency virus (HIV) positive	2	11.8
Others	3	17.6
B. Symptoms		
Cough, fever, loss of weight	4	23.5
Cough and hemoptysis	3	17.7
Cough and fever	4	23.5
Others	6	35.3
C. Chest radiograph (CXR) findings		
Consolidation	9	53.0
Consolidation & cavitation	3	17.6
Consolidation & pleural effusion	2	11.8
Consolidation & lung mass	1	5.8
Others	2	11.8
D. Chest radiograph (CXR) localization		
Right upper lobe (RUL)	5	29.3
RUL & other segments	2	11.8
Right middle lobe (RML)	1	5.8
RML & other segments	3	17.7
Left upper lobe (LUL)	3	17.7
Others	3	17.7

**Table 2 jcm-11-00676-t002:** Bronchoscopic features and subtypes of EBTB.

Bronchoscopic Features	*n*	%
A. Endobronchial Tuberculosis (EBTB) Subtypes		
Caseating	9	53.0
Edematous hyperaemic	3	17.7
Tumorous	2	11.8
Granular	2	11.8
Ulcerative	1	5.8
B. Bronchoscopy localization		
Right upper lobe (RUL )	3	17.7
RUL & other segments	2	11.8
Right Middle Lobe (RML)	2	11.8
RML & other segments	2	11.8
Right Lower Lobe (RLL)	2	11.8
RLL & other segments	2	11.8
Left lobe (Left upper lobe: LUL, Left middle lobe: LML, Left lower lobe: LLL)	4	23.5
C. Endobronchial biopsy HPE		
Caseating Granulomatous inflammation	8	47.0
Granulomatous inflammation	7	41.0
Chronic inflammation	2	12.0
D. Endobronchial biopsy histopathological examination (HPE) with:		
Positive AFB Microscopy	8	47.0
Negative AFB Microscopy	9	53.0
E. BAL AFB Microscopy with:		
Positive AFB Microscopy	6	35.3
Negative AFB Microscopy	11	65.0
F. BAL MTB Culture & Sensitivity (C+S)		
Positive MTB Culture & sensitivity (C+S)	11	65.0
Negative MTB Culture & sensitivity (C+S)	6	35.3

[*n* = Number of cases].

**Table 3 jcm-11-00676-t003:** EBTB Post treatment assessment.

A. Duration of Anti-TB	*n*	%
6 months	10	59.0
7 months	1	5.8
9 months	4	23.6
12 months	1	5.8
2 months	1	5.8
**A. Symptoms improvement post anti-TB treatment**		
Yes	16	94.0
No	1	6.0
**B. Repeat bronchoscopy post anti-TB treatment**		
Yes	6	35.0
No	11	65.0
**C. Bronchoscopy findings post anti-TB treatment**		
Resolved endobronchial lesion	6	35.0
Not available	11	65.0

## Data Availability

The data presented in this study are available on request from the corresponding author. The data are not publicly available due to privacy restriction.

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
