# Peer review of "Bronchoscopic Features and Morphology of Endobronchial Tuberculosis: A Malaysian Tertiary Hospital Experience"

_jcm, 2022, doi:10.3390/jcm11030676_

Round 1
Reviewer 1 Report
I have added comments on the paper.
Please polish your English and see the comments and make the adjustments.

Reviewer 2 Report
This manuscript describes the clinical and bronchoscopy phenotypic features of 17 patients diagnosed of endobronchial tuberculosis between 2018 and 2020. involvement in active pulmonary tuberculosis.
I suggest major revisions to be done. The references included are old-fashioned; this section has to be updated. In addition, I have detected many English composition mistakes and structure weaknesses (i.e., iterations of words in the same paragraphs, verbs), so, if possible, a detailed copyediting must be done.
MAJOR REVISIONS
Abstract
-The results are exposed non-specifically, using vague attributes (commoner, commonest, good clinical response, …), without showing any concrete figures or percentage (just 53% is shown) or a more detailed description of facts. The main features must be written in a similar way as shown in Results section.
-“Commonest” is iterated consecutively four times in the abstract.
Methods
-Line 77: this study is an observational descriptive one, so the adjective descriptive must be added. In addition, this sentence showing the type of study should be in the beginning of the section (the end of the sentence ”[…]in accordance with the Helsinki declaration 2008 and all patients gave their written informed consents.” Must be allocated at the end of section).
-Line 88: Response to treatment were evaluated- were is plural, so the correct verb form is was.
-One of the criteria in the initial population to examine was three negative sputum examinations for acid-fast bacilli (AFB), whereas a selection criterion for the 17 patients was positive AFB-culture on the microbiological examination of bronchoalveolar lavage. This feature can be exposed as a characteristic of the selected patients instead of a selection criterion of initial population. It is explained very well in Discussion.
-Line 91. Two drug combination must be specified.
Results
-Table 1 and Table 2 must be moved to Supplementary materials, and two new tables added with a general description of the characteristics of the cohort (instead of Table 1) and the bronchoscopic characteristics and phenotyping of EBTB (instead of Table 2), with columns showing the frequencies in each case.
-Tables 1 and 2. The footnotes must be alphabetically ordered.
-Line 108. Do not use present perfect tense but past. In general, present cannot be used in Results section (just past or past perfect)
-Line 172 “1 of them”. A number cannot used at the beginning of the sentence. It is better: One of them. The same for 266.
References
-The DOI is included in some articles but not in others.
-The literature is very old, with only two publications from the last five years (numbers 30 and 31).
Reviewer 3 Report
The authors presented a bronchoscopic phenopytic and treatment outcome of endobronchial tuberculosis in a single Malaysian tertiary hospital. Although the topic of this manuscript is interesting, additional descriptions and clarifications are needed.
Major comments
- Lines 43–47
The authors should consider moving these descriptions to the Discussion section.
- Lines 54–56, “Lady Windermere syndrome”
Lady Windermere syndrome is a nodular bronchiectatic nontuberculous mycobacterial lung disease that involves the RML and lingular segment, suggesting that this term is totally unrelated to edonbronchial TB. Thus, this sentence must be removed.
- AFB
1) Some sentences or paragraphs are very confusing as the authors have only used the term “AFB” (e.g., lines 70, 72, 143–145, 200–202). Thus, in these sentences, it is uncertain whether “AFB” means “AFB smear” or “AFB culture” or “AFB smear and culture.” Please use “AFB smear” or “AFB culture” instead of “AFB” alone throughout the manuscript.
2) In addition, a detailed description of the AFB smear and/or culture examination method should be provided in the Methods section. Particularly, was the sputum or samples obtained via bronchoscopy cultured in both solid and liquid media during the study period? If so, what kind of media was used?
- Please confirm whether the TB-PCR test (e.g., Xpert MTB/RIF assay) was conducted for the diagnosis of tuberculosis during the study period. When using the TB-PCR test, the results should be included in the manuscript.
- Lines 69–72
1) Did all the patients with chronic cough and three negative AFB smears and/or persistent consolidation on CXR undergo bronchoscopy during the study period? Considering that this study was retrospectively conducted, it is highly likely that only a small portion of these patients underwent bronchoscopy. Please clarify.
2) It is unclear why pulmonary tuberculosis was suspected to be included in the differential diagnosis in patients with persistent “consolidation.” This is probably because the most common radiographic finding on CXR in adults with pulmonary tuberculosis is cavity or nodules, whereas it could be consolidation in children. Please clarify.
3) Please clearly describe how long the duration in which consolidation is observed on CXR is defined as “persistent.”
- Were there any patients who underwent chest CT for the evaluation of lung infiltration during the study period?
- Line 76, BAL
Please describe BAL in detail in the authors’ center during the study period.
- Table 1, Table 2
1) I believe that Tables 1 and 2 should be presented in a more concise and structured form. Please refer to previous studies (Respirology (2015) doi: 10.1111/resp.12474, Respirology (1997) 2, 275-281) and revise accordingly.
2) “Indication for bronchoscopy” seems to be unnecessary in Table 1 because the authors already described the indication of bronchoscopy in lines 69–72.
3) Please clarify whether “PTB” means AFB smear-negative pulmonary TB or AFB culture-negative pulmonary TB.
4) “Nil” is not commonly used for the description of “absence of underlying disease.” Please use other terms, such as “none.”
5) Furthermore, “cons” is not a commonly used abbreviation. I recommend using the full term “consolidation” rather than just “cons.” The authors actually used “consolidation” in case 11.
- Treatment outcome
1) The authors used “recovery” or “resistance” to describe the outcome of enrolled patients throughout the manuscript (e.g., lines 91–95, 152). However, these terms do not properly describe the treatment outcome of patients with pulmonary tuberculosis. The authors must use the term “cure” or “treatment completion” or treatment success,” according to the definition of the WHO.
2) Please describe the information regarding recurrence after a successful treatment.
- Lines 122 and 123
How can the carina of trachea be evaluated on CXR?
- Lines 135–138
Were there any cases in which the CXR and bronchoscopy findings showed discrepancy to each other?
- Table 3
It appears that 6 out of 17 patients with EBTB underwent repeat bronchoscopy after treatment completion. Was there any indication or other specific reason for the repeat bronchoscopy in these 6 patients?
- The manuscript should be reviewed by an experienced editor whose first language is English and who specializes in editing papers written by scientists whose native language is not English. There are some sentences that are grammatically incorrect (e.g., lines 276–277).
Minor comments
- Line 113
“Recurrent hypoglycemia” is not a symptom of tuberculosis.
- Table 2, Case 6, “LMB”
1) What does “LMB” stand for?
2) Please delete “cons” from the list of abbreviation.
- Line 72
“Between 2018 and 2020” means that the study subjects were enrolled only in 2019. If this is not the case, please use “from 2018 to 2020.”
- Line 114
Smear-negative TB à smear negativity on AFB examination?
Round 2
Reviewer 2 Report
The manuscript has considerably been improved. In addition, all the suggestions performed in my previous review have been done. My only comments left is the correct writing of Mycobacterium tuberculosis. The name of a species has to be written in italics.
Author Response
Reviewer 3 input :
My only comments left is the correct writing of Mycobacterium tuberculosis. The name of a species has to be written in italics.
Response : Agree. have been written
Reviewer 3 Report
Authors have revised the manuscript appropriately according to the comments.
Author Response
Input from Reviewer 3
Authors have revised the manuscript appropriately according to the comments.
Response : Noted with thanks